# Could the Hybridization of the SE/TGfU Pedagogical Models Be an Alternative for Learning Sports and Promoting Health? School Context Study

**DOI:** 10.3390/children10050877

**Published:** 2023-05-13

**Authors:** Ismael López-Lemus, Fernando Del Villar, Amparo Rodríguez-Gutiérrez, Jara González-Silva, Alberto Moreno

**Affiliations:** 1Faculty of Physical Activity and Sports Sciences, University of Extremadura, 10003 Cáceres, Spain; ilopezle@alumnos.unex.es (I.L.-L.); arodriguezgu@unex.es (A.R.-G.); jarags@unex.es (J.G.-S.); 2Centre for Sport Studies, Rey Juan Carlos University, 28942 Madrid, Spain; fernando.delvillar@urjc.es

**Keywords:** hybrid unit, physical education, physical activity, sport education, teaching games for understanding

## Abstract

The present study aims to analyze the influence of the Sport Education (SE)/Teaching for understanding (TGfU) hybrid unit on enjoyment, perceived competence, intention to be physically active, skill execution, decision making, performance and game involvement. A short-term (12-lesson) pre-test/post-test quasi-experimental design was conducted in two groups: control (technical approach: 70 students; age = 14.43 ± 0.693; n = 32 female) and experimental (hybrid unit SE–TGfU: 67 students; age = 13.91 ± 0.900; n = 30 female). The coding instrument was based on the Game performance Assessment Instrument. The Enjoyment and Perceived Competence Scale and the Measure of Intentionality to be Physically Active questionnaire were also used. The results of pairwise comparisons between the groups showed higher post-test scores for most dependent variables for boys and girls using the hybrid SE/TGfU unit. Lower post-test scores were found in pairwise comparisons for several dependent variables in both boys and girls. The present study showed that the application of hybrid models SE/TGfU could increase and help facilitate students’ game involvement and game performance, enjoyment, perceived competence and intention to be physically active, in both boys and girls. In future studies, it would be necessary to analyze psychological variables in the educational context for a deeper assessment.

## 1. Introduction

Research on interventions that support the health and wellness of adolescents has shown that they could generate benefits both now and in their maturity, and even for generations to come [1,2]. Considering this, health and physical activity are closely related [3,4]. Daily routines of physical activity in both children and adolescents have shown various benefits regarding health, physically, socially, psychologically, in terms of growth, and mentally [5,6]. Nevertheless, the World Health Organization (WHO) has pointed out that individuals between 5 and 17 years of age do not follow the daily recommendations of 60 min of moderate-vigorous physical activity per day [7]. Several studies have shown that 80% of the adolescent population does not follow this recommendation [8]. Furthermore, in some cases, girls show lower levels of physical activity than boys [9]. This is more disturbing if the 24-Hour Movement Guidelines [10] are used as a reference. Thus, only 5.4% of the adolescent population achieves the recommended level of sleep, physical activity and screen use, with no differences between genders [11]. The adolescent age is a key moment at which to clear the ground for health, but right at this point, physical activity begins to decrease [12,13].

School centers are an essential environment regarding their promotion of physical activity and healthy habits [14,15,16], and could be the key to a long-term change in health habits [17,18,19]. Physical Education (PE) classes could contribute to increasing health and physical activity levels significantly [20,21,22]. However, there are contradictory results on this point because this effect does not always occur [23,24].

It is necessary to focus on Quality PE (QPE) in order to develop students’ confidence, competence and knowledge in PE classes [3]. This could guarantee healthy and active attitudes towards life in the future, and generate active identities in adolescents [25]. The guidelines published by the United Nations Educational, Scientific and Cultural Organization (UNESCO) point out that the psychomotor, cognitive, social and affective domains are interconnected [26]. Therefore, interventions regarding the quality of PE should attempt to have an impact on these domains [27]. Physical activity alone, “per se”, it is not enough for adolescents to develop and undergo personal growth [28].

On the other hand, the Self-Determination Theory (SDT) postulates that there are three basic psychological needs (BPNs): autonomy, relatedness and competence [29]. The satisfaction of these three BPNs favors a more self-determinate and intrinsic motivation [30]. In PE classes, intrinsic motivation depends, among others, on several factors, such as enjoyment and perceived competence [31]. These factors have been positively linked to a greater possibility of adherence to sport practice [31,32], intention to be physically active [33,34] and improvement in relatedness [35,36]. Enjoyment has been defined as a multidimensional construct [37] that is related to perceptions of competence (mastery and performance), sport-specific activities (team affiliation, competitive excitement), and the influence of social agents (teacher and parental involvement) [38]. For its part, Competence has been defined as the need to have effective interactions with the environment that allow for an improvement in skills and experiences [29].

Various Pedagogical Models (PMs) have emerged in order to address these issues, based on the idea from cross-curriculum pedagogy that learning and teaching are interdependent elements [39]. Their use, both isolated and combined, seems to have a positive effect on four domains: psychomotor, cognitive, social and affective [40,41]. They could provide students with QPE [42]. PMs could be understood as long-term approaches that provide the learning–teaching process with a coherent and comprehensive plan. This allows the teacher to plan effectively in order to achieve specific learning objectives via these plans, actions and decisions, according to both content and context, and attending to all domains [41,42].

However, some factors can influence students’ perception and interfere with the attainment of the benefits of QPE, including gender, age, race, and sociocultural level, among others [43]. In the case of gender, research still shows debatable results [23]. Girls’ involvement and perception of their ability could be influenced by the lesson contents [44], kind of sport [43,45], being in single-gender classes or coeducational environments [46,47], and their social and family context [48]. Even PE teachers tend to create masculinized environments and classes that could cause gender gaps [49].

In order to relieve these effects, PMs promote integrated technical–tactical work, making this the central focus of action, and give prominence to the students and the process [50,51]. Among these PMs, Casey and Kirk (2021) highlight the Sport Education (SE) model [52] and the Game-Centered Approach (GCA) [53] as stand-out exemplars for promoting the adequate teaching of sports in a more equitable and inclusive environment [45,54].

However, employing PMs that are focused solely on the teaching of sports has limitations [55]. To address these issues, Casey and MacPhail (2018) [56] recommended the hybridization of PMs in order to combine the best characteristics of individual models, considering both the individual differences among students and the context [57], and to thus adapt to the current educational framework [50].

Of the two PMs, the SE model provides a greater depth of learning for the duration of the seasons and greater improvements in motivation in terms of group identity. This derives from the adaptation of the institutionalized characteristics of the sport (seasons, affiliations, formal competitions, final events, registrations, and celebrations), in which the students assume different roles. In this way, the SE model favors an improvement in decision making, an increased level of responsibility and autonomy, and developments in students’ physical and social domains [58,59].

The SE model aims to train educated, enthusiastic, and competent athletes who have the ability to perform satisfactory sports practice and to understand and execute strategies that are appropriate to the complexity of the game [52]. However, this requires a higher level of student competence, knowledge, and social skill, and the greater structuring and systematization of tasks [60]. GCAs, such as Teaching Games for Understanding (TGfU), can promote this type of development [61]. The TGfU model prioritizes tactical understanding and actual practice over technical mastery. Additionally, it gives prominence to the students, thus allowing them to become aware of the skills required at different phases of the game, as well as its general structure [62]. By applying the pedagogical principles of simplification, representation, exaggeration, and tactical complexity—addressed through sports modifications at different levels (regulatory, technical, and tactical) and small sided games—TGfU can cultivate an understanding of the game and action in less skilled players [63]. This seems to promote supportive learning and encourage knowledge and competence in sports through play, thus increasing intrinsic motivation and perceived enjoyment [64].

Given the benefits of these individual models, a SE–TGfU hybridization is now widely used [40]. The hybridization of both models could enhance their characteristics [60]. The TGfU model would favor greater systematization and improve social learning, as well as knowledge. On the other hand, the SE would provide a greater depth of learning for the duration of the seasons and improvements in motivation thanks to group identity [40]. Furthermore, both TGfU and SE propose meaningful tasks with defined objectives, supporting students’ involvement. This combination would make it possible to create learning environments that favor the development of all students, regardless of gender, through their experience [40,41].

Experiences of this hybridization have shown that students can improve their autonomy by undertaking responsibilities and decision making, and performing and being involved in real game situations [65,66]. Moreover, this hybridization promotes positive emotions and social interaction by offering opportunities for students to increase their commitment to the group, and this has shown to be independent of gender and content [67]. Nonetheless, little is still known about gender effects [41]. Indeed, some studies indicate that hybridizations may not be suitable for the education of girls in the context of co-education [68].

Both models have inconsistent evidence concerning the different components of game performance [61]. First, there are contradictory results concerning the physical and cognitive domain after the application of the SE model, and this may be influenced by gender or skill level [69,70]. Research applying the TGfU model has shown that the social domain is more strongly influenced by the experiences of students than those of peers [71]. In some cases, studies have shown that gender and initial skill level differences can influence subsequent learning and lead to inequalities, especially at the tactical level [72,73]. It is essential to understand what strategies or content can facilitate gender diversity so that differences in skill are an opportunity, and not a limitation, in learning [47].

Therefore, the current study aimed to compare the influence of two mini-handball didactic units—one using a hybrid SE/TGfU model and the second using a technical approach (TA)—on student enjoyment, perceived competence, intention to be physically active, decision making, skill execution, game performance (GP), and game involvement (GI). We considered gender as a specific variable of interest. It was hypothesized that students, both boys and girls, who were taught using the hybrid unit would similarly improve their scores in decision making, skill execution, GP and GI; consequently, they would report higher levels of enjoyment, perceived competence, and intention to be physically active. Furthermore, we hypothesized that participants who received the TA would show lower levels of all variables, for both boys and girls, compared to the group who experienced the hybrid unit. Consequently, the students who received the TA intervention would maintain or increase the differences between boys and girls both in comparison with their group and with the hybrid group.

## 2. Materials and Methods

### 2.1. Design and Participants

A pre-test/post-test quasi-experimental design was followed. The participants were 137 students (Mage = 14.18; SD = 0.83; n = 62 female) in their second and third year of high school. The groups were divided into two forms of teaching. The groups that were taught handball via a hybrid SE/TGfU were made up of four classes with 70 students in total. The other 67 students from three classes were taught handball via the TA unit. The students in both groups had no prior experience in handball or with SE and TGfU models. This inexperience was the reason for selecting the groups.

The PE teacher was male and had 12 years of high school teaching experience. The PE teacher had experience using SE and TGfU when designing learning tasks. The flexibility of the Spanish curriculum regarding the group of sports that must be developed in each educational phase and the wide experience of the teacher (more than ten years) in teaching handball conditioned the sport content of the units. The research was authorized by a university ethics committee in south-west Spain. The prospective participants and their parents or guardians were informed about the procedures the researchers were going to follow, placing emphasis on the fact that their participation was voluntary. Finally, they signed an assent and consent form.

### 2.2. Procedures and Data Collection

A mini handball (5 × 5) hybrid SE/TGfU and TA units were carried out during twelve sessions in PE classes. The groups, conditions and routines of the school context were not modified, in order to guarantee a natural environment [74]. Sessions were held twice per week, each lasting 55 min.

The PE teachers had often used the TA in the school setting. Although the teacher–researcher had not taught these students previously, he had been working in the high school for two years and was aware of its socio-cultural and pedagogical context. In this sense, the intervention aimed to change this trend in the center and obtain QPE classes.

The PE teacher was a member of the research team and assumed the dual role of teacher–researcher to gain in-depth access to the learning process. The intervention was part of an action–research plan for the teacher (the lead author), who tried to improve the teaching–learning processes in this context. The teacher–researcher had previous experience implementing both SE and TGfU models in different sports and contexts. He had been in contact with the empirical evolution of these pedagogical approaches via his participation in and contribution to conferences, workshops and specific papers. Despite this, it was his first experience teaching a hybrid SE/TGfU model. Therefore, a seminar was developed for the hybridization of both models over a period of three weeks prior to the intervention. During the first week, model hybridization papers (e.g., Gil-Arias et al. [75]) were reviewed to point out the aspects that would facilitate the integration of both SE and TGfU. In the second week, the design of the hybrid unit based on previous research was developed. During the third week, it underwent peer review, with the tasks corrected and redesigned into the fourth week. After this process, the pre-test was carried out.

Pre-test data collection was carried out in the week prior to the intervention over three sessions. The post-test was carried out on real game situations in the last three sessions of the intervention. On the other hand, the questionnaires were completed in the session prior to the intervention (pre-test) and in the session after the intervention (post-test). The teacher–researcher was present during the process to resolve any doubts. The questionnaires had items that were rated on a Likert scale from 1 (strongly disagree) to 5 (strongly agree).

Videotape recordings were captured by two crossed-angled cameras located in mini handball (18 × 13 m) outdoor courts. Two 18 min 5 × 5 games against the same teams were videotaped, with goalkeeper/player changes every 3 min, ensuring a minimum participation of 10 min per player.

#### 2.2.1. Game Performance Assessment Instrument (GPAI)

The coding instrument used for actions during the real game was based on the Game Performance Assessment Instrument (GPAI) [76], which allows the assessment of the “game performance behaviors that demonstrate tactical understanding, as well as the player’s ability to solve problems by selecting, applying appropriate skill” [76] (p. 231), and the measurement of on-the-ball and off-the-ball skills [53].

On-the-ball actions (throwing, passing and bouncing) were coded for Decision making and Skill Execution in attack. Off-the-ball actions (throwing, passing and bouncing mark/guard) were coded in defense for these same items. The components that allowed the assessment to be defined with respect to decision making and execution were included. The decision made was coded either as appropriate (e.g., marking passer or thrower in play) or inappropriate (e.g., marking player out of play), or successful (e.g., intercepting pass or throw) or unsuccessful (e.g., pass or throw reaches the intended target) [58].

The recommendations of Memmert and Harvey [77] were considered for both the calculation of the Decision-Making Index (DMI) (values between 0 and 1) and the Skill Execution Index (SEI). Comparing between the GP and GI variables is necessary in order to determine the real performance of a player [77]. The GP was calculated as (decision made + skill execution) divided by two. The GI was calculated by totaling to the appropriate and inappropriate decision making + appropriate and inappropriate skill execution.

#### 2.2.2. Enjoyment and Perceived Competence

The enjoyment and perceived competence were analyzed with enjoyment and perceived competence scale [78] (ECS), which was adapted to school handball [79,80]. The instrument is made up of seven items, in which three items answer to “enjoyment” (e.g., “I enjoy a lot when I play handball”) and four items answer to “perceived competence” (e.g., “I consider myself very good when I play handball”). Cronbach´s alpha values were confident for “enjoyment” (pre/post-test: 0.92/0.90) and for “perceived competence” (pre/post-test: 0.88/0.86).

#### 2.2.3. Intention to Be Physically Active

The Spanish version of the Measure of Intentionality to be Physically Active (MIFA) [81] was used to analyze students’ intention to be physical active. This scale is made up of five items (e.g., “I usually practice physical activity in my free time”), which begins with the following paragraph: “In relation to your intention to practice any sport or physical activity…”. Cronbach’s alpha values were confident (pre/post-test: 0.86/0.91).

### 2.3. Intervention

#### 2.3.1. Hybrid SE/TGfU Unit Design

The unit was designed according to the principles and features of SE: seasons, affiliation, formal competition, record keeping, final event and festivity [52]. The unit had three phases: (1) a learning phase (lessons 1–4); (2) a formal competition phase (lessons 5–11); and (3) a final event (lesson 12). In the first lesson of the learning phase, and for each class, students were divided by the PE teacher into four persistent teams of six or seven students, after which students developed their team identity (name, image, color and a chant/song). Moreover, and during the learning phase, students experienced different roles (e.g., coaches, journalist, fitness leader, equipment manager), which were determined by the students choosing the role that best fit their personal strengths. Responsibility for the design of the sessions and tasks was progressively integrated depending on the level of the students and their evolution [58].

In the learning phase, the task designed by the PE teacher were set according to four pedagogical principles of the TGfU model: (1) simplification, where a diversified practice in which different problems arise are proposed and time is allowed in order to explore and discover answers (e.g., 3 vs. 3 plus goalkeeper, but only the pivot throws); (2) representation, where games that manipulate the complexity of the formal game to make it simpler and more understandable without distorting the tactical structure are provided (e.g., midfield game without change of possession after losing the mobile in attack); (3) exaggeration, where the secondary aspects of the game, such as spaces, players, sizes or goals, are modified in order to enhance a specific tactical problem (e.g., establishing exclusive defense zones where pressure is eliminated and allows more time in decision making or narrower fields, such as 12 × 18); and (4) tactical complexity, where students are progressively exposed to real game situations that are appropriate to their level of development and evolution (e.g., (a) keeping the ball, (b) progression towards the opponent’s goal, (c) attack–defense transition, (d) occupation of spaces in attack and defense, and (e) organization of attack and defense) [53]. Consequently, the teaching process was developed in a contextualized way, based on modified games that maintained the nature of handball.

Questioning was also used to guide students during the activity [82]. To ensure the search for reflection, attention was focused on “What?”, “Where?”, “When?”, “Why?”, “With whom?” and How?”, and the questions were oriented to categories of time, space and assumed risk [53].

All teams participated in a formal competition before the final event and awards ceremony. The roles adopted by the students enabled them to gain autonomy and independence in this phase. Moreover, the teacher was only a mentor in the final sessions, supporting and helping students according to their needs, and encouraging self-management and providing feedback on their performances [83].

#### 2.3.2. Technical Approach (TA) Unit Design

The lesson design of the technical approach unit was delivered according to the following features: (1) the teacher controlled and made decisions regarding the classes’ contents, design, tasks, interactions and evaluation; (2) practice was evaluated following the desired movements; (3) the task and goals were focused on technical skills development; (4) the teacher was the instructional leader of the unit, who set the learning goals, task progression and pace of work; (5) technical skills were developed in small collaborative and rotary groups; (6) a decontextualized task was used as practice; and (7) the teacher provided prescriptive feedback to correct errors [39].

Nevertheless, the students practiced for as much time as possible in order to enhance their motor involvement and ensure that they had the same number of opportunities.

#### 2.3.3. Interventions Fidelity

To test the influence of a pedagogical model on the dependent variables, it was necessary to validate that the intervention conducted was consistent with the features of the applied models [84]. The first author and one additional observer with experience in pedagogical models in PE randomly selected lessons for the assessment of the presence or absence of the items included in the Instructional Checklist [84] (Table 1).

A sample of four lessons for each pedagogical approach was observed, which was more than 12.5% of the total sample [85]. Here, 100% agreement was reached between the two observers. Therefore, both observers confirmed that all key aspects included in the instructional checklist were applied.

### 2.4. Inter-Observer Reliability

The first author was trained by an expert observer with more than ten years of experience in observational methodology [86]. The training was carried out on more than 20% of the sample in both groups (37 players) during three viewing sessions both pre-test and post-test [87]. An inter-observer reliability of 0.94 (ICC) was reached, and 93.5% agreement was reached for decisions regarding data extraction between the main author and additional observer with experience in observational methodology [88].

### 2.5. Data Analysis

Version 24.0 of the Statistical Package for the Social Sciences (SPSS; IBM Corporation, Armonk, NY, USA, 2016) was used. A preliminary analysis was performed with the Levene and Kolmogorov–Smirnov tests to confirm assumptions regarding the homogeneity of variances and the normality of distribution (*p* < 0.05). The Levene test showed significance (*p* > 0.05) for the variable GI (throw) in the attack action, as well as for the variables DMI (Throw), SEI (Pass), SEI (Bounce) and GP (Pass) in the defense action (Mark/Guard). Given that the Spearman value exceeded 0.70 for the dependent variables in both the pre-test and post-test measurements, the multicollinearity of the variables was assumed [85].

For each group and gender at each of the two different time points (pre-test and post-test), the M and SD were calculated. To compare the between-group and within-group differences, a 2 × 2 × 2 pedagogical model (hybrid model and TA) × test time (pre-test and post-test) × gender (boys and girls) multivariate analysis of variance was conducted. A Bonferroni correction factor was used for these analyses to control for Type 1 errors due to using multivariate comparisons. The effect sizes were calculated using the partial Eta-squared statistic. Effect sizes above 0.01 were considered small, those above 0.06 were considered medium, and those above 0.14 were considered large [89]. The level of statistical significance was established at *p* < 0.05 (95% confidence interval).

## 3. Results

### 3.1. Pre-Test Analysis

In the pre-test, Levene tests were performed to confirm the assumption of the homogeneity of variance (*p* > 0.05). The results showed no significant differences among both groups for all variables considered (enjoyment, *p* = 0.206; perceived competence, *p* = 0.177; intention to be physically active, *p* = 0.658; pass DMI, *p* = 0.593; throw DMI, *p* = 0.621; bounce DMI, *p* = 0.80; pass SEI, *p* = 0.239; throw SEI, *p* = 0.378; bounce SEI, *p* = 0.412; pass GP, *p* = 0.551; throw GP, *p* = 0.427; bounce GP, *p* = 0.193; pass GI, *p* = 0.479; throw GI, *p* = 0.062; bounce GI, *p* = 0.330; pass defense DMI, *p* = 0.288; throw defense DMI, *p* = 0.084; bounce defense DMI, *p* = 0.317; pass defense SEI, *p* = 0.081; throw defense SEI, *p* = 0.203; bounce defense SEI, *p* = 0.162; pass defense GP, *p* = 0.073; throw defense GP, *p* = 0.271; bounce defense GP, *p* = 0.521; pass defense GI, *p* = 0.270; throw defense GI, *p* = 0.391; bounce defense GI, *p* = 472).

### 3.2. Between-Group Post-Intervention Analysis

A significant multivariate effect was found, with a large effect size for both boys (Wilks’ Lambda = 0.213, F(27.000) = 14.634, *p* < 0.001, η_p_^2^ = 0.787) and girls (Wilks’ Lambda = 0.232, F(27.000) = 13.120, *p* < 0.001, η_p_^2^ = 0.768). In the pairwise comparisons, the boys and girls taught using the hybrid TGfU/SE model had significantly higher post-test scores for all the dependent variables when assessed using the GPAI when compared to the boys and girls taught using the TA; this is except for Pass GI and Bounce GI for boys, and Pass GI, Throw GI and Bounce SEI for girls in offensive actions. Significant differences were also not found in Pass defense SEI for boys, and Pass defense GI, Throw defense GI and Bounce defense GI for both boys and girls (see Table 2 and Table 3).

In the same way, the dependent variables of perceived competence and intention to be physically active, for both boys and girls taught using the hybrid TGfU/SE, showed significantly higher post-test scores when compared to boys and girls taught using the TA. On the other hand, the dependent variable of enjoyment did not show significant differences between the groups for both boys and girls (see Table 4).

### 3.3. Within-Group Pre-Post-Intervention Analysis

Within-group multivariate contrasts showed a significant effect with a high effect size for boys (Wilks’ Lambda = 0.190, F(27.000) = 16.885, *p* < 0.001, η_p_^2^ = 0.810) and girls (Wilks’ Lambda = 0.206, F(27.000) = 15.319, *p* < 0.001, η_p_^2^ = 0.794) taught using the hybrid SE/TGfU model. In the pairwise comparison, both boys and girls reported significantly higher values for all the dependent variables when assessed using the GPAI in the post-test compared to the pre-test; this is except for Bounce GI in offensive actions for girls, and Bounce defense GI in defensive actions for both boys and girls.

Moreover, a significant multivariate effect was also found for both boys (Wilks’ Lambda = 0.486, F(27.000) = 4.190, *p* < 0.001, η_p_^2^ = 0.514) and girls (Wilks’ Lambda = 0.561, F(27.000) = 3.103, *p* < 0.001, η_p_^2^ = 0.439) taught using the TA unit. In the pairwise comparison, the girls had significantly lower post-test scores for Throw DMI, Bounce DMI, Pass GP and Bounce GI, and both boys and girls had significantly lower scores for Pass SEI, Bounce SEI, Bounce GP and Pass GI in offensive actions. For defensive actions, girls had significantly lower post-test scores in Bounce defense DMI and Throw defense SEI, while boys in Bounce defense SEI and Bounce defense GP, as well as both boys and girls in Pass defense SEI, Pass defense GP, Pass defense GI and Bounce defense GI, had significantly lower post-test scores (see Table 2 and Table 3).

On the other hand, in the pairwise comparisons, both groups, TGfU/SE and TA, showed significantly higher post-test scores for enjoyment and perceived competence for both boys and girls. However, intention to be physically active showed significantly lower values for boys taught using the TA unit, and did not show any significant values for girls. Meanwhile, the group taught using the TGfU/SE unit showed significantly higher values for both boys and girls in the post-test scores (see Table 4).

## 4. Discussion

The current study aimed to evaluate the impact of a hybrid SE/TGfU didactic unit on students’ enjoyment, perceived competence, intention to be physically active, decision making, skill execution, performance, and game involvement. Specifically, it also aimed to test whether there were differences between boys and girls with similar skill levels. Our first hypothesis was that students, both boys and girls, who were taught using the hybrid unit would similarly improve their scores in decision making, skill execution, GP and GI, and consequently have higher levels of enjoyment, perceived competence and the intention to be physically active. Furthermore, we hypothesized that participants who received the TA would show lower levels of all variables, for both boys and girls, compared to the group who experienced the hybrid unit.

The dependent variable of enjoyment showed significantly higher values for students taught under the SE/TGfU unit. These results endorse what the scientific literature had previously established for both the isolated use of TGfU [90] and SE [59], or their hybridizations [67,91]. Nevertheless, the group taught usingthe TA, both for boys and girls, also showed higher values for enjoyment. This could be due to low initial enjoyment values [67]. However, in this case, the initial values were already high. It seems that enjoyment in PE classes could be affected by other factors regardless of the model applied. Factors such as class content, motivational climate or the teacher’s attitude, among others, could be considered [23,92]. Regarding enjoyment, despite there being no initial differences between the groups, both for boys and girls, the improvements in the values did not provide significant differences between the groups. This contrasts with previous research in which differences between the groups were found [75,91,93].

The perceived competence values were found to be significantly higher for the group taught using the hybrid SE/TGfU unit than those for the group taught using the TA unit, during both the within-group and between-group comparisons. This improvement has been linked to QPE experiences [94], which could promote the continuity of physical activity in the future [15,16]. However, the TA group showed a significative improvement from the initial values. This fact confirms what some studies suggest about the acquisition of technical skills in early sport learning [17]. An improvement in this variable could even influence the student’s enjoyment, because it could help to create a more meaningful learning experience [14]. Therefore, this perceived competence should be compared with students’ performance and competence in actual game situations [18]. In addition, a defined analysis of the offensive and defensive game actions should be also carried out [19].

In terms of the offensive actions, both boys and girls taught with the hybrid SE/TGfU didactic unit recorded higher scores regarding decision making and skill execution. Moreover, there were no gender differences for this effect. Students in this hybrid group also improved in terms of GP for all attack actions, again without gender differences, which implies that they had a better global understanding of the game. These results corroborate those of prior works on hybrid SE/TGfU in which gender was considered [18,20,21,22], and together indicate that there is a need to adapt tasks to the educational stage of the students, independent of gender and skill level.

Memmert and Harvey [23] suggested that GP indices should be prioritized for age groups in which the reflective parts of the session can have a strong impact, but that it should always be compared with GI. Only one prior study on hybrid SE/TGfU has considered this issue [22]. In this study, both boys and girls in the hybrid unit group improved in terms of GI for most of the variables considered; the exception is the dribble in attack, for which there was an increase for boys and a decrease for girls. One possibility is that this reflects a better use of the dribble action by boys due to improved decision making and skill execution, for which GP is significant. In turn, girls used the dribble action less as an offensive resource, but more effectively, resulting in an improvement in both GI and GP.

From the present study, it is evident that the number of actions and interventions increases (GI) alongside game performance (GP). However, it remains necessary to assess whether this increase is produced by the greater continuity of the game or by an increase in actions at the same time as participation. This issue has been investigated in the context of the SE and TGfU models separately [24], but not in hybridizations. Providing equal opportunities for practice, and insisting that the teacher ensures the equitable and balanced participation of all students, regardless of sex, is the key to the improvement of all students and the promotion of co-education [21]. Such an increase in participation can come from the choice of modifications and tasks, as smaller games in terms of the number of players and space encourage interaction. Therefore, in the case of handball, opting for 5 vs. 5 and 4 vs. 4 structures, for example, in these stages of education, is likely to encourage participation [25].

Compared to the students that were taught using the hybrid unit, those taught using the technical approach showed improvements in only some of the attack variables. Both boys and girls showed a significant improvement in the use of the throwing action to attack, both in terms of GP and GI. Similarly, both boys and girls showed a higher number of shots and a better performance, although this improvement in both sexes was only obtained in execution, not decision making (boys). In addition, the mean values indicate that the students taught using the technical approach made better decisions regarding the pass action. A significant increase in GP and GI was also obtained for the pass; that is, students made more passes with better performance. In the case of the dribble, there were almost no improvements, and there was a decrease in the average values. Specifically, both boys and girls showed significant decreases regarding decision making for this action, and GI only significantly improved for boys.

For students taught using the technical approach, the results showed less consistent improvements, and in some cases, there were clear differences between boys and girls. Moreover, for some variables, there was no evidence of an improvement in either sex. For example, while both boys and girls showed numerical improvements in decision making when defending against passes and shots, the improvement in defending against shots was only significant for girls. While the execution and GP decreased, GI showed increases for both sexes. These results imply that students preferred this action in attack, meaning that there was a need to defend on more occasions, but also that they did not defend more effectively.

The defensive actions of boys and girls in the hybrid unit group improved significantly. However, in the case of defending the dribble, there was no improvement in GI for either sex; that is, the mean number of dribble actions before and after the intervention was similar.

Our first hypothesis was that students, both boys and girls, who were taught using the hybrid unit would similarly see improvements in their decision-making, skill execution, GP and GI scores, and consequently have higher levels of enjoyment, perceived competence, and intentions to be physically active. Furthermore, we hypothesized that participants who were taught using the TA would show lower levels of all variables, for both boys and girls, compared to the group who were taught using the hybrid unit. The results partly confirm support for this hypothesis. This is because boys and girls taught using the hybrid unit show high levels of decision making, skill execution, GP and GI, and consequently have higher levels of enjoyment, perceived competence, and intentions to be physically active. These results agree with previous studies of both the SE and TGfU models separately and of their hybridizations, in which students show better results than when taught using the TA [95,96,97,98]. Enjoyment and perceived competence have been positively linked to a greater possibility of adherence to sport practice, which is similar to the intention to be physically active [31,32]. Developing students’ confidence and competence in PE classes could guarantee healthy and active attitudes to life in the future, generating active identities in adolescents [25]. However, on the other hand, the group taught using the TA did not show lower levels in all variables. In fact, some variables had improved. Previous studies of using the TA have suggested that it could enable students to improve their performance and skill level [99]. The baseline is very important for these improvements and teachers have to pay attention because initial skill level differences can influence subsequent learning [72]. In this way, the inexperience of the students regarding the handball content could be the reason for these improvements. Nevertheless, in the TA group, both boys and girls show better results for competence and enjoyment, but do not show this in the intention to be physically active. This differs from previous studies, in which competence and enjoyment showed a positive link.

Furthermore, we hypothesized that participants who were taught using the TA would show lower levels of all variables, for both boys and girls, compared to the group who were taught using the hybrid unit. Consequently, the students who were taught using the TA intervention would maintain or increase the differences between boys and girls, both in comparison with their group and with the hybrid group.

When comparing the groups after the intervention, it was evident that the boys and girls taught using the hybrid model generally improved in decision making, skill execution, and game performance for all game actions relative to the students taught using the technical approach. However, for some variables, these differences were not significant for both sexes, namely GI for the passing action, GI for pass defense, GI for shot defense, and GI for dribble defense. In these cases, the related GP variables did show differences, which seems to indicate a similar increase in the number of actions in both groups, but with effectiveness only improving for boys and girls in the hybrid SE/TGfU group.

Regarding attack actions, all students slightly increased their number of passes, although this increase was larger, and indeed significant, for boys and girls who were taught using the hybrid model. Despite the reduced space and improved defensive actions, these students were more effective in attacking when using the pass. Moreover, even though the boys did not significantly increase their number of dribble actions in attack, and even though the girls did not significantly increase their number of shots, the performance of these actions did improve. Hence, the results seem to indicate that these students had a better understanding of the game and of the indicated moments at which to enact their ball skills during attack. Further, these actions were defended with greater skill, resulting in greater effectiveness in the game.

The case of defense actions is very striking, where, observing the means, it can be seen how the number of actions is similar or greater in the case of the hybridization group, although this is not significant between groups. However, in comparison, it is evident that the students in the hybrid group had a higher GP regarding these actions compared to those taught using the technical approach. One possibility is that the reduced space situations enhanced the use of defensive actions over offensive actions. 

It is noteworthy that for the hybrid model, girls improved in a similar way to boys, while the girls taught using the technical approach did not present the same improvement. The results imply that it is highly relevant to consider students’ prior skill and knowledge of the sport and ensure that they are given opportunities to participate in tasks that are appropriate to their needs [18,22]. In addition, the results indicate that increases in game participation, or a greater number of game actions, do not necessarily translate into game performance. Therefore, tasks must ensure the resolution of real tactical problems regarding the actions of later play [20].

The intention to be physically active showed significantly higher values for the group taught using the SE/TGfU model compared to the group taught using the TA unit, both for boys and girls. On the other hand, the group taught using the TA unit showed significantly lower values after the intervention, for both genders. In fact, the comparison between the groups shows significant differences in the SE/TGfU group, both for boys and girls. These results are similar to previous studies [9,26]. Therefore, the hybridization of pedagogical models could contribute to generating future active identities in adolescents [27,28]. Nevertheless, other studies point out that methodological change is not enough in order to guarantee healthy periods of physical activity in PE [29]. PE should educate active citizens in order to maintain an adequate level of physical activity beyond the school context [30,31]. The adequate planning of tasks, and keeping the specific skills required of the contents in mind, is essential [34]. In this way, the need to generate QPE classes remains, because we cannot load the full weight of this responsibility onto the subject of PE [32].

Although the results may indicate the positive effect of the hybrid model on the performance, involvement, skill level, perceived competence, enjoyment and intention to be physically active of the students, the results should be taken with caution because it is a preliminary study with a small sample. Consequently, future research could extend the current sample. In addition, we used a quasi-experimental approach in which we examined changes in class groups already established in a natural context. Nevertheless, we conducted an ecologically valid intervention [74]. However, in this sense, the SE/TGfU unit presented in this study was evaluated, keeping in mind the PE class in the Spanish context, where units are usually limited to 8–10 lessons. Future studies should extend the length and duration of the hybrid unit according to the suggested length of an SE season [52]. Perhaps, units of 15–20 lessons could provide similar results, and could minimize the baseline effect and the influence of the inexperience of the students regarding the content of the sport [95].

The examination of additional variables beyond performance and skill execution could be desirable in order to reinforce knowledge regarding the intention to be physically active. On this point, BPN satisfaction could point towards the better quality of PE [33]. It would be necessary to analyze the BPN satisfaction and the environmental influences, as well as the actual practice of the sport during extracurricular time [35].

## 5. Conclusions

The present study shows that improvements in the students’ game performance and game involvement after being taught using a hybrid SE/TGfU model were similar for both boys and girls, despite this model having a short duration and occurring in a natural co-educational environment. The results imply that it is essential to understand the specific needs of each game action in order to correctly design tasks that favor learning. As such, correctly defining the requirements of each sport would help teachers to apply this model more effectively and generate a better environment in order to improve the quality of physical education. In this case, it seems that the model promoted the students’ intention to be physically active, enjoyment and perceived competence. However, it is important to acknowledge the limitations of this study. The effects of the learning process throughout the annual program has not been assessed, and in the same way, the influence of different schools’ community agents was not considered.

Likewise, the application of these hybrid models can help facilitate the greater involvement of students in the game. Such involvement is enhanced both in quantity and quality, or with a better understanding of the game and the effectiveness of the relevant sports skills. These improvements could favor the quality of PE classes and promote more active and healthy behaviors. However, there is a need to further assess whether hybrid units influence the psychological variables of students in order to understand their wider impact in the educational context. Moreover, the fact that the baseline effect must be considered in future research should be kept in mind. Nevertheless, although this study delves into a differentiated analysis between sexes, the gender should be considered as a variable of interest. This would help to determine the actual individual needs of each student in a sport learning context, and make it easier for teachers to plan according to the personal needs and abilities of their students.

## Figures and Tables

**Table 1 children-10-00877-t001:** Instructional Checklist.

Date	Present	Absent
1. Group of students go to a designated home area and begin warming up with that group.		
2. Students warm up as a whole class under the direction of the teacher.		
3. Performance records are kept by students		
4. All the tasks are related to the small sided game that is being taught.		
5. Students practice individually or in small groups under the direction of the teacher.		
6. Students perform specialized roles within their group/team.		
7. Modifications to the full game are performed.		
8. Student performance scores count towards a formal and public scoring system.		
9. Tasks designed by the teacher are highly structured and based on the repetition of technical skills.		
10. Students practice in a decontextualized context		
11. Student success criteria are based on the successful execution of technical skills		
12. Students are employed for at least 30 min in the practice of modified games.		

Note: items 1, 3, 5, 6 show features from the SE model, items 2, 5, 6, 10, 11 are from the TA, and items 4, 7, 12 are from the TGfU.

**Table 2 children-10-00877-t002:** Descriptive statistics, between-group post-intervention and within-group pre–post-intervention analysis of each dependent variable (offensive actions: pass, throw and bounce).

		Pre-Intervention Hybrid SE/TGfU Unit	Post-Intervention Hybrid SE/TGfU Unit			Pre-Intervention Direct Instruction Unit	Post-Intervention Direct Instruction Unit		
Variables	Gender	*M* (*SD*)	*M* (*SD*)	*p*	95% CI	*M* (*SD*)	*M* (*SD*)	*p*	95% CI
Pass DMI	Boys	50.21 (3.92)	58.24 (6.10)	<0.001 ^a^	[−0.098, −0.062]	50.30 (4.24)	53.92 (2.83)	<0.001 ^d^	[−0.055, −0.018]
	Girls	47.72 (4.41)	56.16 (5.54)	<0.001 ^a^	[−0.104, −0.065]	50.27 (5.80)	53.10 (2.81)	0.007 ^c^	[−0.049, −0.008]
Throw DMI	Boys	50.39 (2.93)	57.24 (3.26)	<0.001 ^a^	[−0.081, −0.055]	48.81 (2.99)	50.92 (3.07)	0.002 ^d^	[−0.034, −0.008]
	Girls	50.12 (3.01)	58.25 (3.60)	<0.001 ^a^	[−0.095, −0.067]	49.43 (2.34)	50.60 (3.86)	0.117 ^d^	[−0.026, −0.003]
Bounce DMI	Boys	49.50 (3.65)	54.97 (3.76)	<0.001 ^a^	[−0.068, −0.042]	52.46 (5.41)	50.51 (4.04)	0.004 ^d^	[0.006, 0.033]
	Girls	50.72 (4.63)	54.00 (4.18)	<0.001 ^a^	[−0.047, −0.019]	51.83 (4.60)	50.50 (4.36)	0.072 ^c^	[−0.001, 0.028]
Pass SEI	Boys	50.89 (4.12)	58.08 (5.82)	<0.001 ^a^	[−0.093, −0.051]	50.38 (4.36)	51.35 (4.47)	0.375 ^d^	[−0.031, 0.012]
	Girls	49.16 (3.90)	56.72 (5.06)	<0.001 ^a^	[−0.099, −0.052]	50.67 (6.04)	51.80 (4.19)	0.352 ^d^	[−0.035, 0.013]
Throw SEI	Boys	49.76 (3.28)	56.87 (5.08)	<0.001 ^a^	[−0.089, −0.053]	49.22 (3.84)	52.08 (3.20)	0.002 ^d^	[−0.047, −0.011]
	Girls	50.06 (3.72)	57.06 (5.02)	<0.001 ^a^	[−0.089, −0.051]	48.67 (2.86)	50.90 (4.57)	0.028 ^d^	[−0.042, −0.002]
Bounce SEI	Boys	48.11 (3.55)	54.97 (5.10)	<0.001 ^a^	[−0.085, −0.043]	50.27 (3.85)	51.08 (4.68)	0.323 ^d^	[−0.024, 0.008]
	Girls	49.62 (3.07)	53.50 (3.50)	<0.001 ^a^	[−0.056, −0.021]	50.40 (3.30)	51.93 (3.63)	0.093 ^a^	[−0.033, 0.003]
Pass GP	Boys	50.55 (3.83)	58.08 (5.76)	<0.001 ^a^	[−0.094, −0.057]	50.30 (4.07)	52.59 (3.45)	0.018 ^d^	[−0.042, −0.004]
	Girls	48.47 (3.84)	56.50 (4.93)	<0.001 ^a^	[−0.101, −0.060]	50.43 (5.73)	52.33 (3.23)	0.077 ^d^	[−0.040, 0.002]
Throw GP	Boys	50.08 (2.40)	57.00 (3.44)	<0.001 ^a^	[−0.082, −0.056]	49.03 (2.90)	51.46 (2.92)	<0.001 ^d^	[−0.037, −0.011]
	Girls	50.09 (2.63)	57.59 (3.22)	<0.001 ^a^	[−0.089, −0.061]	49.56 (2.69)	50.73 (4.08)	0.021 ^d^	[−0.031, −0.003]
Bounce GP	Boys	48.82 (2.99)	54.82 (3.93)	<0.001 ^a^	[−0.072, −0.048]	51.32 (4.18)	50.78 (3.38)	0.365 ^d^	[−0.006, 0.017]
	Girls	50.19 (3.51)	53.69 (3.49)	<0.001 ^a^	[−0.048, −0.022]	51.10 (3.67)	51.13 (3.17)	0.960 ^c^	[−0.013, 0.013]
Pass GI	Boys	16.42 (4.60)	21.42 (9.31)	<0.001 ^a^	[−7.278, −2.722]	17.78 (5.63)	18.76 (3.21)	0.406 ^a^	[−3.281, 1.335]
	Girls	15.00 (4.49)	18.00 (6.53)	0.018 ^a^	[−5.482, −0.518]	15.20 (5.03)	17.40 (2.88)	0.092 ^a^	[−4.763, 0.363]
Throw GI	Boys	9.74 (3.09)	11.26 (3.73)	<0.001 ^a^	[−2.323, −0.729]	8.78 (1.89)	9.62 (2.62)	0.042 ^b^	[−1.645, 0.030]
	Girls	9.41 (2.09)	11.81 (3.27)	<0.001 ^a^	[−3.275, −1.538]	9.40 (2.88)	10.40 (4.56)	0.029 ^a^	[−1.897, −0.103]
Bounce GI	Boys	6.71 (4.94)	7.42 (6.16)	0.030 ^a^	[−1.351, −0.070]	8.27 (5.25)	9.03 (5.35)	0.023 ^a^	[−1.406, −0.108]
	Girls	5.53 (3.68)	5.00 (4.03)	0.135 ^a^	[−0.167, −1.229]	8.07 (5.04)	8.67 (5.16) ^d^	0.102 ^d^	[−1.321, 0.121]

Note. M = Mean; SD = Standard Deviation; IC = Confidence Interval. Between-group post-intervention analysis is reported with superscripts (a, a = *p* > 0.05; a, b = *p* < 0.05; a, c = *p* < 0.01; a, d = *p* < 0.001).

**Table 3 children-10-00877-t003:** Descriptive statistics, between-group post-intervention and within-group pre–post-intervention analysis of each dependent variable (defensive actions against pass, throw and bounce).

		Pre-Intervention Hybrid SE/TGfU Unit	Post-Intervention Hybrid SE/TGfU Unit			Pre-Intervention Direct Instruction Unit	Post-Intervention Direct Instruction Unit		
Variables	Gender	*M* (*SD*)	*M* (*SD*)	*p*	95% CI	*M* (*SD*)	*M* (*SD*)	*p*	95% CI
Pass Defense DMI	Boys	52.24 (5.50)	57.92 (7.13) ^a^	<0.001 ^a^	[−0.081, −0.032]	50.65 (4.62)	50.51 (4.42)	0.002 ^d^	[−0.023, 0.026]
	Girls	52.87 (4.58)	58.03 (7.24) ^a^	<0.001 ^a^	[−0.078, −0.025]	48.93 (6.05)	49.13 (3.72)	0.003 ^d^	[−0.029, 0.025]
Throw Defense DMI	Boys	49.34 (4.74)	57.92 (2.61) ^a^	<0.001 ^a^	[−0.101, −0.070]	49.78 (3.43)	47.32 (3.50)	0.002 ^d^	[0.009, 0.040]
	Girls	51.06 (3.52)	56.75 (2.68) ^a^	<0.001 ^a^	[−0.074, −0.040]	49.70 (3.08)	47.00 (4.36)	0.003 ^d^	[0.010, 0.044]
Bounce Defense DMI	Boys	50.58 (3.27)	55.13 (2.96) ^a^	<0.001 ^a^	[−0.058, −0.033]	49.27 (2.69)	47.92 (2.49)	0.042 ^d^	[0.000, 0.047]
	Girls	49.91 (2.32)	55.91 (3.01) ^a^	<0.001 ^a^	[−0.074., −0.046]	49.23 (2.80)	48.70 (3.31)	0.468 ^d^	[−0.009, 0.020]
Pass Defense SEI	Boys	46.95 (8.45)	54.03 (7.18) ^a^	<0.001 ^a^	[−0.104, −0.037]	49.11 (8.73)	50.57 (8.06)	0.399 ^a^	[−0.049, 0.020]
	Girls	49.13 (6.14)	55.88 (8.28) ^a^	<0.001 ^a^	[−0.074, −0.046]	48.73 (6.36)	49.60 (6.86)	0.652 ^c^	[−0.074, 0.029]
Throw Defense SEI	Boys	48.79 (5.20)	54.53 (5.36) ^a^	<0.001 ^a^	[−0.079, −0.035]	48.14 (4.12)	45.76 (4.58)	0.036 ^d^	[0.002, 0.046]
	Girls	47.56 (5.60)	54.09 (5.15) ^a^	<0.001 ^a^	[−0.089, −0.041]	47.57 (5.41)	46.03 (5.63)	0.220 ^d^	[−0.009, 0.040]
Bounce Defense SEI	Boys	50.71 (3.00)	54.37 (4.55) ^a^	<0.001 ^a^	[−0.057, −0.017]	48.78 (3.36)	47.95 (3.88)	0.417 ^d^	[−0.012, 0.029]
	Girls	49.97 (3.76)	54.16 (5.30) ^a^	<0.001 ^a^	[−0.064, −0.020]	50.00 (5.13)	46.97 (4.59)	0.009 ^d^	[0.008, 0.053]
Pass Defense GP	Boys	49.66 (4.37)	55.95 (6.02) ^a^	<0.001 ^a^	[−0.084, −0.042]	49.84 (4.74)	50.57 (4.57)	0.498 ^d^	[−0.029, 0.014]
	Girls	50.94 (3.51)	56.91 (6.53) ^a^	<0.001 ^a^	[−0.083, −0.037]	48.67 (3.22)	49.33 (3.58)	0.577 ^d^	[−0.030, 0.017]
Variables	Gender	*M* (*SD*)	*M* (*SD*)	*p*	95% CI	*M* (*SD*)	*M* (*SD*)	*p*	95% CI
Throw Defense GP	Boys	49.13 (3.39)	56.18 (3.32) ^a^	<0.001 ^a^	[−0.084, −0.057]	49.00 (2.40)	46.57 (2.98)	<0.001 ^d^	[0.011, 0.038]
	Girls	49.37 (2.92)	55.34 (2.73) ^a^	<0.001 ^a^	[−0.074, −0.045]	48.67 (3.02)	46.57 (4.66)	0.006 ^d^	[0.006, 0.036]
Bounce Defense GP	Boys	50.61 (2.24)	54.68 (3.07) ^a^	<0.001 ^a^	[−0.053, −0.028]	49.03 (2.14)	47.97 (2.24)	0.104 ^d^	[−0.002, 0.023]
	Girls	49.94 (1.93)	55.03 (3.57) ^a^	<0.001 ^a^	[−0.065, −0.037]	49.63 (2.44)	47.83 (2.93)	0.013 ^d^	[0.004, 0.032]
Pass Defense GI	Boys	15.58 (4.39)	19.84 (6.19) ^a^	<0.001 ^a^	[−6.385, −2.141]	16.92 (5.30)	18.43 (4.00)	0.166 ^a^	[−3.664, 0.637]
	Girls	15.12 (5.47)	18.93 (5.47) ^a^	0.001 ^a^	[−6.125, -1.500]	17.23 (4.32)	18.80 (3.39)	0.170 ^a^	[−4.055, 0.721]
Throw Defense GI	Boys	9.61 (2.65)	11.00 (3.53) ^a^	0.001 ^a^	[−2.196, −0.594]	9.81 (2.36)	11.29 (3.41)	<0.001 ^a^	[−2.298, −0.675]
	Girls	8.50 (2.38)	10.06 (3.31) ^a^	0.001 ^a^	[−2.435, −0.690]	9.20 (2.09)	10.53 (3.19)	0.004 ^a^	[−2.235, −0.432]
Bounce Defense GI	Boys	7.13 (2.35)	7.63 (3.28) ^a^	0.132 ^a^	[−1.136, 0.136]	7.05 (2.13)	7.24 (2.68)	0.562 ^a^	[−0.834, 0.455]
	Girls	7.34 (1.77)	7.88 (2.32) ^a^	0.122 ^a^	[−1.224, 0.162]	7.10 (1.92)	7.53 (2.39)	0.233 ^a^	[−1.149, 0.282]

Note. M = Mean; SD = Standard Deviation; IC = Confidence Interval. Between-group post-intervention analysis is reported with superscripts (a, a = *p* > 0.05; a, b = *p* < 0.05; a, c = *p* < 0.01; a, d = *p* < 0.001).

**Table 4 children-10-00877-t004:** Descriptive statistics, between-group post-intervention and within-group pre–post-intervention analysis of each dependent variable from ECS and MIFA.

		Pre-Intervention Hybrid SE/TGfU Unit	Post-Intervention Hybrid SE/TGfU Unit			Pre-Intervention Technical Approach Unit	Post-Intervention Technical Approach Unit		
Variables	Sex	*M* (*SD*)	*M* (*SD*)	*p*	95% CI	*M* (*SD*)	*M* (*SD*)	*p*	95% CI
Enjoyment	Boys	3.78 (0.98)	4.43 (0.66)	0.002 ^a^	[−1.071, −0.245]	3.24 (1.29)	4.05 (0.91)	<0.001 ^a^	[−1.229, −0.392]
	Girls	3.23 (1.02)	4.07 (0.83)	<0.001 ^a^	[−1.284, −0.383]	3.12 (1.08)	3.73 (1.15)	<0.001 ^a^	[−1.076, −0.146]
Perceived competence	Boys	2.83 (1.05)	3.52 (0.77)	<0.001 ^a^	[−1.018, −0.337]	2.34 (0.87)	3.02 (0.85)	<0.001 ^b^	[−1.021, −0.331]
	Girls	2.15 (0.88)	3.42 (0.73)	<0.001 ^a^	[−1.637, −0.895]	2.05 (0.85)	2.85 (1.04)	<0.001 ^b^	[−1.183, −0.417]
Intention to be physically active	Boys	3.62 (1.15)	4.47 (0.54)	<0.001 ^a^	[−1.183, −0.522]	3.79 (1.02)	3.29 (1.02)	0.004 ^d^	[0.168, 0.838]
	Girls	3.41 (1.01)	4.39 (0.45)	<0.001 ^a^	[−1.348, −0.627]	3.21 (0.98)	3.15 (1.15)	0.724 ^d^	[−0.305, 0.439]

Note. M = Mean; SD = Standard Deviation; IC = Confidence Interval. Between-group post-intervention analysis is reported with superscripts (a, a = *p* > 0.05; a, b = *p* < 0.05; a, c = *p* < 0.01; a, d = *p* < 0.001).

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
