# Peer review of "Could the Hybridization of the SE/TGfU Pedagogical Models Be an Alternative for Learning Sports and Promoting Health? School Context Study"

_children, 2023, doi:10.3390/children10050877_

Round 1

Reviewer 1 Report

The study “Could hybridization of pedagogical models be an alternative 2 for learning sports and health? School context study” examined the effects of a hybrid model of Sport Education (SE) / Teaching Games for understanding (TGfU) on students’ handball skills and motivational variables.

 This is an interesting study. However, authors should consider some issues highlighted below to increase clarity and improve the quality of their manuscript.  

 Considering that this paper focus on a hybrid model of Sport Education (SE) / Teaching Games for understanding (TGfU), this should also be reflected of the title of the manuscript.

Similarly, in the title of the study the term “health” is used but this is not so clear how is reflected the content of the manuscript itself. Have authors examined the effects of their intervention in students’ learning regarding health? Which are the outcome variables related to learning health used in this study. Authors should clarify all these issues and they may consider editing the title of their manuscript.

 There are a lot of double spaces within the texts. Please edit

 p. 69: authors may provide some examples of these Pedagogical Models (PM) to help readers to understand what they mean

 p.102: please provide the full writing of the TGfU and after that use the abbreviation

 A strong rational of using a hybrid SE/TGfU model should be provided in the introduction. Authors should explain, based on both theoretical and empirical evidence, their choice to test this hybrid model.

 Is there any particular reason that authors selected the handball for applying their hybrid model? A justification is needed.

Moreover, authors stated that students did have any experience with handball. Is not handball included in physical education curriculum in Spain? Authors should clarify this issue adding some more details about the settings of their study and particularly about school physical education settings in Spain.

 The hypotheses of this study should be presented and justified.

 Authors stated that “participants were 137 students”, but next in the same paragraph they also stated that “70 students from four classes and 67 students from three classes were taught handball via a hybrid SE/TGfU and TA unit, respectively”. Please clarify the number of participants in this study.

 Please clarify if the same teacher taught both the experimental and the control classes (i.e., Technical approach unit). Considering that the teacher deliver the intervention was also part of the research team and thus he/she was not blind to the research aims, authors should provide a strong rational for their choice and to comment about how possible this may be a confound for the validity of their study. 

 Similarly, to the above issue, the first author (that is the teacher who implemented the intervention) involved in checking the fidelity of the intervention. This raise severe questions regarding the validity of the results. Authors should reconsider this choice.

 A rational for involving gender as a factor in the analyses should be provided. Considering gender had doubled the analyses conducted and made difficult for reader to follow the results. What this approach has added to this paper?

 Authors should check for differences in baseline between two groups and report the respective results.

 Authors should provide and discuss the practical implications of the results of their study

 The limitations of this study should be further discussed.

 Texts should be checked for errors in English grammar and syntax.

 A large number of references were cited (i.e., 107). Authors may consider reducing this number

Texts should be checked for errors in English grammar and syntax.

Author Response

Dear  Reviewer,

The first of all is express our gratitude for taking into consideration our work with minor revisions. 

We submit a review of our paper trying to answer to your comments and suggestions. We grandly appreciate detailed reviews of our paper that allow us to improve the quality of the original paper

We are going to try to explain the changes made bellow. We have followed always the considerations pointed out by you. Likewise, we express our gratitude in order to comments and assessments made, just as giving your time to it.

- The study “Could hybridization of pedagogical models be an alternative 2 for learning sports and health? School context study” examined the effects of a hybrid model of Sport Education (SE) / Teaching Games for understanding (TGfU) on students’ handball skills and motivational variables.

This is an interesting study. However, authors should consider some issues highlighted below to increase clarity and improve the quality of their manuscript.  

 We express our gratitude for you in order to assessment made, just as giving your time to it. We will try to explains all the changes made in order to take in consideration all your suggestions and comments.

- Considering that this paper focus on a hybrid model of Sport Education (SE) / Teaching Games for understanding (TGfU), this should also be reflected of the title of the manuscript.

We have changed the title accordingly. Thank you for your assessment.

- Similarly, in the title of the study the term “health” is used but this is not so clear how is reflected the content of the manuscript itself. Have authors examined the effects of their intervention in students’ learning regarding health? Which are the outcome variables related to learning health used in this study. Authors should clarify all these issues and they may consider editing the title of their manuscript.

We have changed the title accordingly. On the other hand, maybe we have made a syntax mistake in the title. We try to explain the implication of the intervention for the health through psychological variables such as perceived competence and enjoyment. In this sense, we use the “Intention to be physically active” variable and we connect it both the psychological variables and performance variables. We don´t try to explain any variable related to learning health. We are very sorry . Thank you very much for your accurate assessment

- There are a lot of double spaces within the texts. Please edit

Checked out them and edited

- p. 69: authors may provide some examples of these Pedagogical Models (PM) to help readers to understand what they mean

We have redefined the concept of pedagogical models concept on this line.

- p.102: please provide the full writing of the TGfU and after that use the abbreviation

Thank you very much for your assessment. We have provided the full writing.

- A strong rational of using a hybrid SE/TGfU model should be provided in the introduction. Authors should explain, based on both theoretical and empirical evidence, their choice to test this hybrid model.

We have added a new explanation about the benefits and suggestions of this kind of hybridizations, such as SE/TGfU. Why can these models complement each other in learning sports? We try to answer this question in p. 117-124 with this paragraph:

“The hybridization of both models could enhance their characteristics [60]. TGfU model would favor greater systematization and improve social learning, as well as knowledge. On the other hand, the SE would provide greater depth in learning for the duration of the seasons and improvements in motivation for group identity [40]. Furthermore, both TGfU and SE propose meaningful tasks with defined objectives, supporting students involvement This combination would make it possible to create learning environments that favor the development of all students, regardless of gender, through their experience [40,41].”

Thank you very much for your comments

- Is there any particular reason that authors selected the handball for applying their hybrid model? A justification is needed.

The inexperience of the students was the reason. We will clarify this reason in p. 161-162 with this sentence:

This inexperience was the reason in order to select the groups.”

- Moreover, authors stated that students did have any experience with handball. Is not handball included in physical education curriculum in Spain? Authors should clarify this issue adding some more details about the settings of their study and particularly about school physical education settings in Spain.

The Spanish curriculum allows the teacher to select the contents to freely develop the curriculum objectives. The curriculum only makes a description of the groups of sports that will be developed in each educational phase, and teacher must follow these recommendations in his teacher plan. We will try to explain this idea in p. 165-167 with this paragraph:

The flexibility of the Spanish curriculum regarding the set of the group of sports that must be develop in each educational phase and the widely experience in teaching handball of the teacher (more than ten years) conditioned the sport content of the units

- The hypotheses of this study should be presented and justified.

The hypotheses have been added both introduction and discussion. The Hypotheses was showed in the introduction on this way:

“It was hypothesized that students, both boys and girls, who was taught under the hybrid unit would similarly improve their scores in decision making, skill execution, GP and GI, and consequently higher levels of enjoyment, perceived competence, intention to be physically active. Furthermore, we hypothesized that participants who received the TA would show lower levels of all variables, for both boys and girls, compared to the group who experienced the hybrid unit. Consequently, the students who received de TA intervention would maintain or increase the differences between boys and girls both in comparison with their group and with the hybrid group.”

These hypotheses will be discussed in the pertinent section

- Authors stated that “participants were 137 students”, but next in the same paragraph they also stated that “70 students from four classes and 67 students from three classes were taught handball via a hybrid SE/TGfU and TA unit, respectively”. Please clarify the number of participants in this study.

We have rewritten the sentence in order to clarify this aspect. The numbers are correct, but the way of expressing it could be confusing. Thank you very much for your suggestion. This is the new way:

“Participants were 137 students (Mage = 14.18; SD = .83; n = 62 female) and they were in their second and third year of high school. The groups were divided into two units of teaching. The groups that were taught handball via a hybrid SE/TGfU were made up of four classes with 70 students in all. Other 67 students from three classes were taught handball via TA unit.”

- Please clarify if the same teacher taught both the experimental and the control classes (i.e., Technical approach unit). Considering that the teacher deliver the intervention was also part of the research team and thus he/she was not blind to the research aims, authors should provide a strong rational for their choice and to comment about how possible this may be a confound for the validity of their study. Similarly, to the above issue, the first author (that is the teacher who implemented the intervention) involved in checking the fidelity of the intervention. This raise severe questions regarding the validity of the results. Authors should reconsider this choice.

The intervention was part of an action-research. On this kind of project proposal, the teacher use to be part of the research team because is looking for a change in his context.

Trying to keep his influence isolated, we explain both in intervention fidelity and interobserver reliability that there is always another research team mate cooperating. This another research member made an assessment of the interventions and reliability of the data extraction.

We will try to emphasize these appreciations in order to guarantee the validity of the both intervention and results.

We have added examples in several points, like p.177-185 (action-research), p 286-295 (interventions fidelity) and p.299 – (interobserver reliability)

Thank you very much for your suggestion and comments

 - A rational for involving gender as a factor in the analyses should be provided. Considering gender had doubled the analyses conducted and made difficult for reader to follow the results. What this approach has added to this paper?

Thank you very much for your suggestion. The reason was provided in the introduction, between p. 125 -140. But we have try to be more clarify in this sense both into discussion and conclusion, adding practical implications for coeducational contexts that avoiding gender differences and allow both girls and boys to achieve the units objectives in the same way.

In order to make easier for reader to follow the results we will provide another way to show the tables

 - Authors should check for differences in baseline between two groups and report the respective results.

Thank you so much. We have included in the document a section entitled "Pretest Analysis" with this information. 

- Authors should provide and discuss the practical implications of the results of their study.

- The limitations of this study should be further discussed

These points have been added in the discussion. Thank you very much for your suggestion

- Texts should be checked for errors in English grammar and syntax.

Checked out them and edited. Thank you very much for your assessment.

 - A large number of references were cited (i.e., 107). Authors may consider reducing this number.

Some references have been removed

Reviewer 2 Report

I would like to thank to authors for this interesting quality study. I believe that there is a merit for this journal. This study has many strengths  including tests and measurement and reliability and validity as well. Literature reviews is very detailed and up to the point. My only suggestions is about conclusion part. I believe that conclusion part should be very strong and must include practical suggestions for the future studies. Thank you.

Author Response

We express our gratitude for you in order to comments and assessments made, just as giving your time to it. In this sense, we have increased both the discussion and the conclusion trying to clarify and show the practical suggestions that the PE teachers could draw from this study. Again, thank you very much.
